# Optimization of Polyphenol Extraction with Potential Application as Natural Food Preservatives from Brazilian Amazonian Species *Dalbergia monetaria* and *Croton cajucara*

Vaneska Aimee Paranhos de Araújo [1,2], Jefferson Romáryo Duarte da Luz [2], Naikita Suellen da Silva e Silva [2],
Matheus Pereira Pereira [2], Jardel Pinto Barbosa [3], Darlan Coutinho dos Santos [4], Jorge A. López [2],
Lilian Grace da Silva Solon [1] and Gabriel Araujo-Silva [2,*]

1   Post-Graduation Program in Pharmaceutical Sciences, Federal University of Amapá, Rodovia Josmar Chaves
    Pinto Km 02, Jardim Marco Zero, Macapá 68903-419, AP, Brazil
2   Organic Chemistry and Biochemistry Laboratory, State University of Amapá (UEAP), Av. Presidente Vargas,
    s/n-Centro, Macapá 68900-070, AP, Brazil
3   Laboratory of Computational Chemistry, State University of Amapá (UEAP), Av. Presidente Vargas,
    s/n-Centro, Macapá 68900-070, AP, Brazil
4   College of Natural Sciences, State University of Amapá (UEAP), Av. Presidente Vargas, s/n-Centro,
    Macapá 68900-070, AP, Brazil
*   Correspondence: gabriel.silva@ueap.edu.br; Tel.: +55-84-99660-9327

**Abstract:** Scientific interest has currently focused on natural products as a feasible approach for new food additives to replace synthetic preservatives. Thereby, the objective of this work was to optimize the polyphenol extraction from native Amazonian plant species *Dalbergia monetaria* L.f. and *Croton cajucara* Benth., and they were determined by the total content of these compounds. Accordingly, the hydroalcoholic extract's phytocomposition was analyzed by ultra-high-performance liquid chromatography–diode array detector (UPLC-DAD) and various assays to determine the antioxidant capacity (e.g., 2,2-Diphenyl-1-picrylhydrazyl (DPPH) free radical scavenging, ferric reduction power, peroxidation inhibition). In addition, response surface methodology applying a central composite design was used to optimize the antioxidant compound extraction conditions. Extract phytochemical profiles identified polyphenols such as (-)-epigallocatechin gallate, rutin, and hyperoside in both species. Furthermore, *D. monetaria* and *C. cajucara* extracts displayed significant antioxidant capacity, exhibiting similar values compared to the standard synthetic antioxidant butylated hydroxytoluene. Nevertheless, *C. cajucara* showed more antioxidant efficiency compared to *D. monetaria*. These results were consistent with the distribution matrix obtained by a Central Composite Design since the *C. cajucara* extracts exhibited the best response to the adopted optimization model. Therefore, data are promising for obtaining potential options for natural additives for the food industry.

**Keywords:** natural additives; Amazon; *Dalbergia monetaria*; *Croton cajucara*





## 1. Introduction

Nowadays, synthetic preservatives play a key role in preserving food, becoming the most widespread industrial method worldwide [1]. Based on their low cost and ease of use, these compounds have become an integral part of industrial food processing to preserve and minimize foodstuffs deterioration without organoleptic alteration at all supply chain stages [2,3]. However, despite the substantial importance of additives in preventing food oxidation, one of the stages of its deterioration scientific evidence shows the adverse effects of synthetic preservatives, harmfully impacting human health [4–6]. In this concern, the circular economy stimulates studies to develop the next generations of additives based on natural bioactive compounds aimed at food safety, quality, and shelf-life extension [7–9].

Therefore, these studies have concentrated efforts on searching for natural food preservatives as a renewable and sustainable alternative to meet market and consumer demands

for natural additives instead of conventional synthetic compounds, like butylated hydroxytoluene and butylated hydroxyanisole [10–12]. On this basis, natural extracts as a secondary metabolite reservoir have aroused scientific interest in prospecting compounds aimed at developing innovative products or formulations with the potential industrial application since these extracts are Generally Recognized as Safe (GRAS) [13]. Furthermore, although their most widespread use is as spices and pharmacological applications [14], in recent years, these extracts have also been highlighted in the food industry by the bioactive compound incorporation in foodstuffs and active packaging by providing antioxidant and antimicrobial functions for enhancing food quality [7,15,16].

In this current trend, studies have focused on evaluating the phytochemical reservoir of plant biodiversity, highlighting polyphenols for their high antioxidant capacity besides their established beneficial effect on human health. Thus, compounds have been widely described considering several pharmacological activities for the treatment of diseases closely associated with oxidative stress [17,18]. Based on this ability, phenolic compounds deserve great attention from the food sector for their critical role as free radical scavengers, metal ion chelants, prevention of lipid peroxidation, and antimicrobial effect [19–21]. Hence, these bioactive compounds, either as extracts or isolated molecules, are a research target due to their potential biotechnological applications as additives in fresh or processed foods (dyes and preservatives), chemical inputs, and packaging development [8,22,23].

In the context of this trend towards natural antioxidants and other bioactive compounds, Amazonian biodiversity stands out for its potential contribution to the bioeconomy as a sustainable and efficient strategy to prospect phytochemical resources with high industrial added value [24]. Despite its technological potential, only a few studies have addressed the development of new products applying this phytocomposition as food preservatives. Instead, studies have focused mainly on pharmaceutical applications based on popular knowledge for primary health care as a subsidy to validate uses and drug discovery [25–27]. This prospecting is also a strategy to value the Amazonian biodiversity, which is part of the rational use of natural resources, aiming to provide data for their preservation, considering the constant degradation of this biome by anthropic action [28,29].

Based on the vast Amazonian biodiversity of plant species, *Dalbergia monetaria* L.f. (Leguminosae family) and *Croton cajucara* Benth. (Euforbiaceae family) were selected considering ethnobotanical and phytochemical studies. Both species are widely distributed in the Amazon region with significant medicinal use by traditional communities to treat symptoms or diseases involving oxidative stress due to their anti-inflammatory, antiulcerogenic, and antimicrobial activities [26,30]. Ethnobotanical data analyzed by the use of report indices indicated the popular consumption of these endemic species, providing pivotal information for screening their chemical diversity [26,30,31]. Overall, this approach offers a viable alternative regarding secondary metabolites and their relationship with sustainable economic activities and species conservation [25].

*D. monetaria*, popularly known as "verônica," is a typical shrub from Amazon estuary floodplains in Amapá and Pará states, also presenting an area of transition with the plateau of Guiana. Their stem bark and leaves are used in folk medicine to treat human diseases related to oxidative stress effects [31,32]. Although data regarding the chemical composition are still scarce, phytochemical studies have reported a significant phenolic compound and flavonoid content isolated from stem bark and leaves [31–34] and proanthocyanidins as the main bark secondary metabolite [32,35], achieving a promising species due to its significant antioxidant activity. Overall, these compounds display a range of biological effects, such as antioxidant and antibacterial effects, besides other biological activities, such as antineoplastic and antiparasitic [26,32,34].

*C. cajucara* is another widely distributed in the eastern and central region of the Brazilian Amazon rainforest, commonly found in the Amapá and Pará states. Named "sacaca" by the native population, this plant has a historic safe use in folk medicine [30]. The leaves and stem bark of this tree are traditionally consumed as an infusion or decoction, or pills since ethnopharmacological information indicates its use for primary health care to treat different

human disorders associated with oxidative stress [30,36]. Concerning the phytochemical analysis, the stem bark and leaf extracts have revealed significant amounts of chemical compounds, such as simple phenolic compounds, flavonoids, cajucarinolide, alkaloids, sitosterol-3-*O*-β-glucoside, and terpenes [37–39]. These compounds are recognized for having a strong antioxidant capacity [40]. Furthermore, studies indicate that this phytocomposition is closely related to other biological activities, such as anti-inflammatory, antimicrobial, antitumor, antimicrobial, and antinociceptive effects [30,38,39].

The extracts obtained from Amazon rainforest plant matrices are exploited and marketed for medicinal purposes based on their biological activities. Nonetheless, most of these extracts are prepared and supplied without any quality control and scientific validation of their attributed properties [26,31]. Therefore, the potential of plant biomass to obtain inputs rich in biomolecules and its application in several industrial sectors requires an in-depth study aimed at its safe use, the feasibility of its commercialization, and predatory extractivism minimization [23,41]. At this point, it is noteworthy that the global natural extracts market, including also the related segment of the global natural antioxidants market, was valued at USD 11.94 billion in 2021 with a forecast of USD 25 billion by 2030 due to the annual growth rate of 8.9% for the period 2022–2030 [42]. Hence, this growth trend offers opportunities that stimulate the market for extracts and natural antioxidants considering the current challenges of the food industry for safety, quality, and shelf-life [43,44].

Based on the natural product role as effective antioxidant agents in food production and preservation, this study reports for the first time the use of *D. monetaria* and *C. cajucara* extracts, plant species from the Amazon rainforest, as promising food additives. Hence, the objective of this work was to evaluate the effects of the biomass amount and the ethanol concentration to optimize the polyphenol extraction process by applying the Response Surface Methodology and to analyze these extracts regarding their antioxidant capacity and phytochemical composition to determine the optimal extractive condition.

## 2. Materials and Methods

### 2.1. Materials and Reagents

Thiobarbituric acid, trichloroacetic acid, 1,1-diphenyl-1-pycrylhydrazyl, 2,2′-azobis-2-amidinopropane (AAPH), 2,2-Diphenyl-1-picrylhydrazyl (DPPH), butylated hydroxytoluene (BHT) and references standards (gallic acid, chlorogenic acid, catechin, epigallocatechin, rutin, hyperin, quercetin, apigenin, and kaempferol) were supplied by Sigma-Aldrich Co. (St. Louis, MO, USA). HPLC-grade acetonitrile and methanol, and Folin-Ciocalteau reagent were purchased by Vetec (São Paulo, Brazil), and 0.22 μm syringe filters were obtained from Allcrom (São Paulo, Brazil). All other chemicals and reagents were of analytical grade.

### 2.2. Botanical Material and Extract Preparation

*D. monetaria* bark and *C. cajucara* leaves were collected in a floodplain area in the Porto Grande municipality, State of Amapá, Brazil. These species were identified taxonomically by Dr. Tony David Santiago Medeiros (Institute for Scientific and Technological Research of the State of Amapá), depositing the respective specimens as 02. II. 2018, R.D.C. Araujo, 3 HAMAB and 02. II. 2018, R.D.C. Araujo, 2 HAMAB at the Herbário Amapaense (Macapá, AP, Brazil). After cleaning, both species' aerial parts were individually air-dried at 40 °C for 48 h. *D. monetaria* bark and *C. cajucara* leaves were crushed into smaller pieces using a knife mill to a particle size < 180 μm before submitting individual samples to ethanol extraction at 100%, 70% and 40% (*w/v*) at room temperature during seven days. The component extraction by maceration process was performed in the following ratios 1:10, 1:20, and 1:40 (*w/v*) of powdered dried material:solvent. Then, samples were subjected to gravimetric analysis to determine the optimal extraction condition.

### 2.3. Gravimetric Analysis

This analysis was performed by individually transferring 1mL of each sample to crucibles followed by heating for 24 h at 105 °C during a period ranging from 1 to 7 days,

according to the minimum observed variation in the relative standard deviation (SD). This analysis was performed as a quantitative approach to determine the best extraction condition for target compounds.

### 2.4. Experimental Design

The extraction parameters for *D. monetaria* and *C. cajucara* biomasses were optimized using a Response Surface Methodology (RSM). The solvent selected for the extraction was ethanol based on an experimental design adopted due to the effects of the response of each test performed regarding the variation of the assumed factors (Table 1). The combined effect of the variables was evaluated by a factorial experimental design with two factors and three levels ($2^3$). The controlled variables (factors) were: (A). sample mass in the solvent under the following ratios: 10/1 ($-1$), 20/1 (0) and 40/1 (1), and (B). ethanol (extractor solvent) at 40% ($-1$), 70% (0), and 100% (1). Other factors, such as sample particle size (<180 μm), temperature (room temperature), and shelter of the light, were kept constant.

**Table 1.** Factorial planning and determination variables under study for the optimization of the two main variables involved the sample mass:solvent ratio (SM and ethanol concentration (%EtOH) as extractor solvent). First column: number of standard runs; second column: repetition number; third column: experimental blocks; fourth and fifth columns: factorial design results for each variable under study according to the adopted experimental model (two factors and three levels; $2^3$).

| Variable Coded Values | | | | | Variable Coded Values (Continuation) | | | | |
|---|---|---|---|---|---|---|---|---|---|
| Standard Run | Replicate | Block | SB | %EtOH | Standard Run | Replicate | Block | SB | %EtOH |
| 17 | 2 | 3 | 0.025 | 100 | 1 | 1 | 1 | 0.050 | 100 |
| 4 | 1 | 2 | 0.025 | 70 | 7 | 1 | 3 | 0.100 | 40 |
| 5 | 1 | 2 | 0.100 | 100 | 20 | 3 | 1 | 0.100 | 70 |
| 10 | 2 | 1 | 0.050 | 100 | 9 | 1 | 3 | 0.050 | 70 |
| 15 | 2 | 2 | 0.050 | 40 | 12 | 2 | 1 | 0.025 | 40 |
| 16 | 2 | 3 | 0.100 | 40 | 13 | 2 | 2 | 0.025 | 70 |
| 11 | 2 | 1 | 0.100 | 70 | 8 | 1 | 3 | 0.025 | 100 |
| 24 | 3 | 2 | 0.050 | 40 | 19 | 3 | 1 | 0.050 | 100 |
| 6 | 1 | 2 | 0.050 | 40 | 3 | 1 | 1 | 0.025 | 40 |
| 18 | 2 | 3 | 0.050 | 70 | 14 | 2 | 2 | 0.100 | 100 |
| 26 | 3 | 3 | 0.0250 | 100 | 27 | 3 | 3 | 0.050 | 70 |
| 2 | 1 | 1 | 0.100 | 70 | 23 | 3 | 2 | 0.100 | 100 |
| 22 | 3 | 2 | 0.025 | 70 | 21 | 3 | 1 | 0.025 | 40 |
| 25 | 3 | 3 | 0.100 | 40 | | | | | |

### 2.5. Total Flavonoids Content

The total flavonoid content was determined in triplicate according to the aluminum chloride colorimetric assay described by Wannes et al. [45]. Briefly, a 250 μL extract sample (50 μg/mL) was mixed with 2.5 mL of 5% aluminum chloride and 0.5 mL of 1M NaOH, adjusting with distilled water to a final volume of 2.5 mL. After incubation for 30 min in the dark at room temperature, absorbance was read against a blank at 420 nm (BEL UV-M51 Spectrophotometer, China). The total flavonoid content, as determined from a quercetin calibration curve, ranged from 5–30 μg/L ($R^2$ = 0.9978), expressing the results in mg of quercetin equivalents per g of extract.

### 2.6. Total Phenolic Content

The total phenolics content was performed in triplicate by the Folin-Ciocalteau method [46]. Shortly, an ethanolic extract sample aliquot (0.5 mL) was mixed with 0.125 mL of the Folin-Ciocalteau reagent. After 5 min, 2 mL of 7.5% sodium carbonate was added, and the final volume was adjusted to 3 mL with distilled water, mixing the reaction system vigorously. Afterward, the reaction was incubated (room temperature/2 h), and then the absorbance was measured against a blank at 760 nm (BEL UV-M51 Spectrophotometer,

China). The extract's total phenolic contents were calculated by a gallic acid calibration curve (50–200 mg/L, $R^2 = 0.999$). Data were expressed as the mean of mg equivalent of gallic acid per gram of extract.

### 2.7. DPPH Free Radical Scavenging

The DPPH free radical scavenging assay was performed in triplicate as described by Brand-Williams et al. [47], using BHT as the standard positive control. Aliquots of 0.1 mL of extract sample (12.5 μg/mL) were mixed with 3.9 mL of 60 μM DPPH. After vigorous stirring, the reaction mixture was left in the dark for 30 min before measuring the absorbance at 512 nm, using 60 μM DPPH as a blank (BEL UV-M51 Spectrophotometer, China). Free radical scavenging (*FRS*) was calculated using the equation $FRS\ (\%) = 100 \times \left( \frac{Ac - As}{Ac} \right)$, where $A_C$ is the absorbance of the control (all reagents except the target compound) and $A_S$ is the absorbance of the tested substance. Results were expressed as the mean and standard deviation of the 50% minimum inhibitory concentration (IC50).

### 2.8. Total Antioxidant Capacity by Phosphomolybdenum Assay

Total antioxidant activity was determined in triplicate according to the phosphomolybdenum method described by Prieto et al. [48]. An aliquot of 0.1 mL of extract (200 μg/mL) was added to a vial containing 1 mL of reagent solution (0.6 M sulfuric acid, 28 mM sodium phosphate, and 4 mM ammonium molybdate). After capped vials incubation (95 °C/90 min), samples were cooled to room temperature, and the absorbance was measured at 695 nm against a blank (reagents free of target compound) (BEL UV-M51 Spectrophotometer, China). Total antioxidant activity was calculated from gallic acid and BHT calibration curves (2.5–20 mg/L). Results were expressed as μmol equivalent of gallic acid or BHT per gram of extract.

### 2.9. Ferric Reduction Power Assay

The analysis of the ferric-reducing power of *C. cajucara* and *D. monetaria* extracts was performed according to the methodology described by Santos et al. [49]. Briefly, an aliquot of extract sample (100 μg/mL) was mixed with 2.5 mL of 0.2 M phosphate buffer (pH 6.6) and 2.5 mL of 1% (*w/v*) ferrocyanide and the reaction mixture incubated at 45 °C for 20 min. Then, 2.5 mL of 10% trichloroacetic acid was added to the reaction system. After shaking, 2.5 mL of this solution was transferred to another tube, followed by the addition of 2.5 mL of distilled water and 0.5 mL of 0.1% $FeCl_3$ under stirring. Absorbance was determined at 700 nm against a blank (BEL UV-M51 Spectrophotometer, China). All determinations were performed in triplicate, using BHT (100%) as a control standard under the same reaction conditions.

### 2.10. Ultra High-Performance Liquid Chromatography-Diode Array Detection (UPLC-DAD)

*C. cajucara* and *D. monetaria* extracts were analyzed by UPLC-DAD according to García-Villalba et al. [50], using extract solutions at a concentration of 5 mg/mL dissolved in methanol and filtered through a 0.22 μm membrane. Only extracts corresponding to the best response condition according to the adopted experimental model were evaluated. Analyzes were performed using a UPLC system (LC-2030C 3D Plus, Shimadzu, Japan) equipped with a vacuum degasser, autosampler, quaternary pump, and a diode array detector (DAD). The injection sample volume was 2 μL, and chromatographic separation was conducted on a Shimadzu C-18 reverse phase column (100 × 4.6 mm, particle size 2.1 μm) using a gradient solvent system of 1% formic acid (A) and acetonitrile (B) as mobile phases at a flow rate of 1 mL/min. Determinations were achieved in triplicate at 30 °C for 30 min, applying a gradient system starting at 5% B and ramped up to 95% B for fingerprint analysis. Both identification and quantification were made using external calibration (gallic acid, chlorogenic acid, catechin, epigallocatechin, rutin, hyperin, quercetin, apigenin, and kaempferol). Peaks were identified by comparing the retention time at 280 nm for single phenolic compounds by a standard curve for gallic acid (0.1–100 μg/mL, y = 0.0032x + 134.56, $R^2 = 0.9999$), while flavonoids were detected at 340 nm using a curve for quercetin

(0.5–500 µg/mL, y = 0.0047x + 256.79, $R^2$ = 0.9989). Data were expressed as mg of the corresponding standard/g of extract.

### 2.11. Lipid Peroxidation Inhibition Assay

The lipid peroxidation assay was applied to quantify thiobarbituric acid reactive species (TBARS), according to the adapted procedure described by Melo et al. [51], using an egg yolk homogenate as a lipid-rich substrate. Shortly, egg yolks (10 g) were homogenized by sonication in 90 mL of 20 mM PBS (pH 7.32). Afterward, an aliquot of the homogenate (1.75 mL) was mixed with 0.05 mL of extract at a concentration of 25, 50, or 100 µg/mL. BHT was used as a reference antioxidant. After inducing lipid peroxidation with the addition of 0.2 mL of 0.12 M AAPH, the reaction system was incubated for 30 min at 37 °C and cooled before adding 4 mL of 0.7% thiobarbituric acid solution in 15% trichloroacetic acid. The reaction was again incubated (100 °C/45 min), cooled, and 3 mL of n-butanol was added to eliminate any non-malondialdehyde interference. After centrifugation ($2000 \times g$/5 min), the supernatant was used to monitor lipid peroxidation at 532 nm (BEL UV-M51 Spectrophotometer, China). Data were expressed as µg equivalent of TBARS per gram of extract.

### 2.12. Statistical Analysis

Results of the antioxidant extraction optimization were treated by STATISTICA® software for Windows for surface response tests (RSM), Pareto charts, desirability, and predicted and observed values [52]. Data on lipid peroxidation (mean ± SD) were analyzed by ANOVA followed by Tukey-Kramer's post hoc test using GraphPad Prism Software version 7.0 (San Diego, CA, USA). Statistical significance was set at $p < 0.05$ [53].

## 3. Results and Discussion

### 3.1. Optimization of Polyphenol Extraction by Response Surface Methodology and Characterization by Antioxidant Capacity

Plant matrices as a source of natural products have become an attractive market since extracts obtained from these biomasses are a rich reservoir of bioactive compounds (e.g., phenolic compounds, flavonoids) [11,17,41]. These compounds are important for the development of food, nutraceutical, cosmetic, and pharmaceutical products due to consumer awareness and changing habits associated with knowledge regarding the harmful impact of synthetic substances on health [43,44].

Despite the chemical composition relevance, mainly polyphenols as the major contributors to the antioxidant activity, their extraction represents a challenge since this procedure plays a pivotal role in obtaining bioactives from plant biomass [54,55]. The extractive procedure can affect the target compounds that are liable to degradation under the action of factors such as solvent, pH, temperature and even the plant biomass amount, which achieves the yield and quality of the final product [56]. Therefore, the appropriate extraction method and its factors are relevant aspects of selecting and optimizing the extractive production conditions [57]. Thereby, several studies show efforts to verify the effect of several methods to develop and compare extraction processes to recover better bioactive yields from plant matrices due to their high added value for the industrial sector [41,58,59].

Therefore, a statistical design applying response surface methodologies can be an alternative to optimize the extractive process conditions for obtaining high-value-added secondary metabolites [60]. In order to improve the recovery of antioxidant compounds from *D. monetaria* and *C. cajucara* biomass, the combined effect of controlled variables was conducted by selecting the amount of biomass and different ethanol concentrations. Ethanol was selected as the extractor solvent since polar biologically active compounds are extracted with hydroethanolic solution, as well as to decrease any solvent interference in the polyphenol extraction that can affect its antioxidant capacity [61]. Thereby, conditions referring to the established variables of biomass and ethanol proportions were evaluated for each test using response graphs, subjecting the adopted factors to a Box–Behken-type factorial analysis with two factors and three levels ($2^3$), as shown in Table 1.

The interactive effects between variables for obtaining phenolic compounds from biomasses were analyzed based on the response surface, using a response surface plot and Pareto diagrams. In addition, the optimization of the extraction process was analytically monitored through the determination of the total contents of phenolic compounds and flavonoids of each extract obtained as established by the statistical design to verify the influence of the related variables on the extractive procedure. Figures 1 and 2 depict the relationship between the independent and dependent variables, graphically represented by the 3D response surface, while the Pareto diagrams indicate significant interactions between the variables for the recovery of both phenolic compounds and flavonoids from plant biomasses of the species *D. monetaria* and *C. cajucara*, respectively.

Overall, the Pareto diagrams referring to the quantification of total phenolics displayed no significance regarding the %EtOH applied to extract these compounds from *D. monetaria* biomass (Figure 1). Nevertheless, the mass:solvent ratio was significant in the polyphenol extraction using *C. cajucara* biomass samples ($p < 0.05$; Figure 2). Concerning total flavonoids, the % EtOH variable was significant in the extraction procedure using *D. monetaria* samples, while the mass amount and the volume of solvent were significant for extractions with *C. cajucara* biomass, as depicted in Figures 1 and 2, respectively.

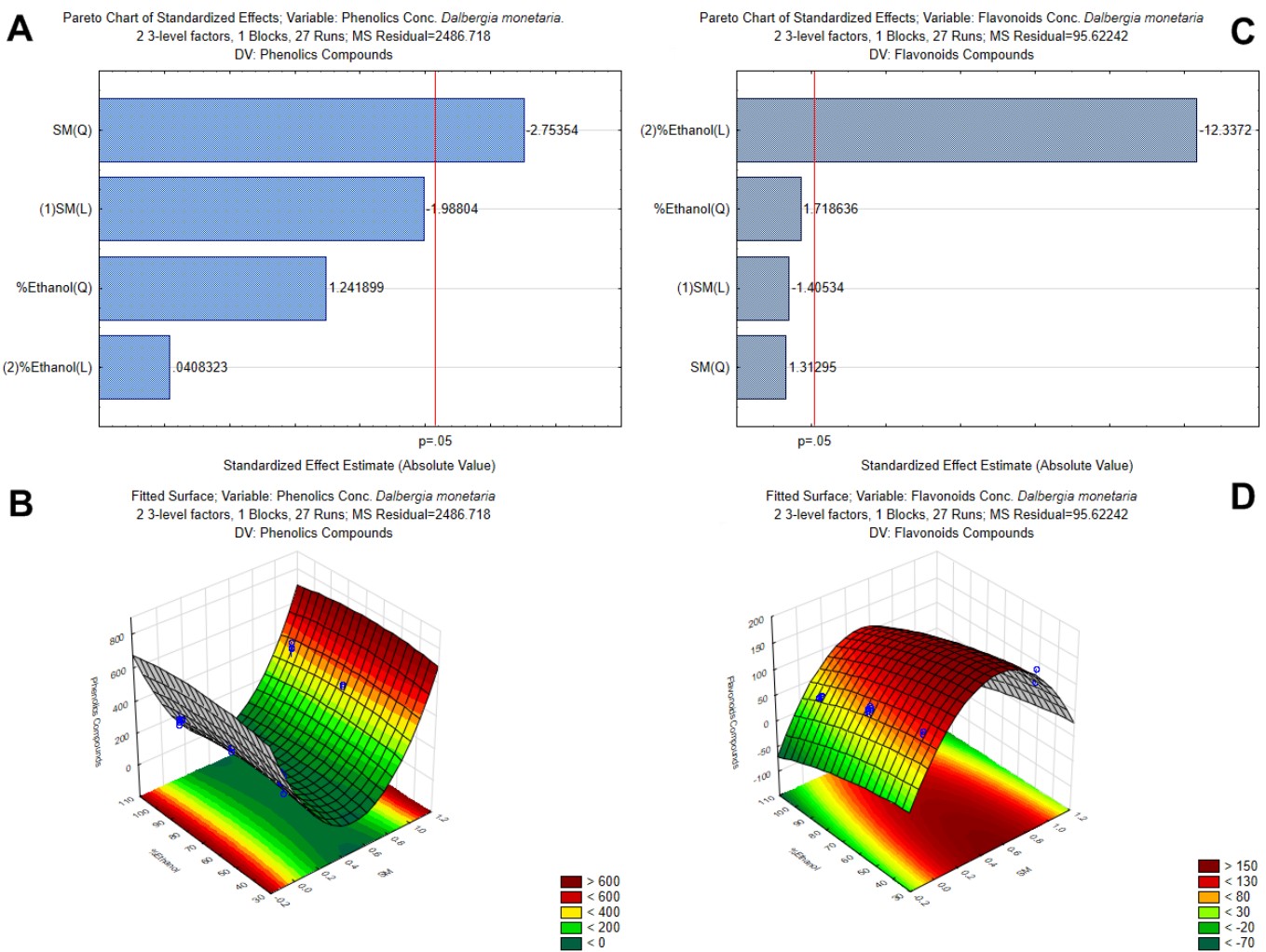

**Figure 1.** Surface response plots and Pareto diagrams showing the effect of the variables amount of plant biomass of *D. monetaria* and concentration of ethanol as solvent extractor (mass:solvent ratio) to evaluate the content of phenolic compounds (**A**,**B**) and flavonoids in the obtained extracts (**C**,**D**).

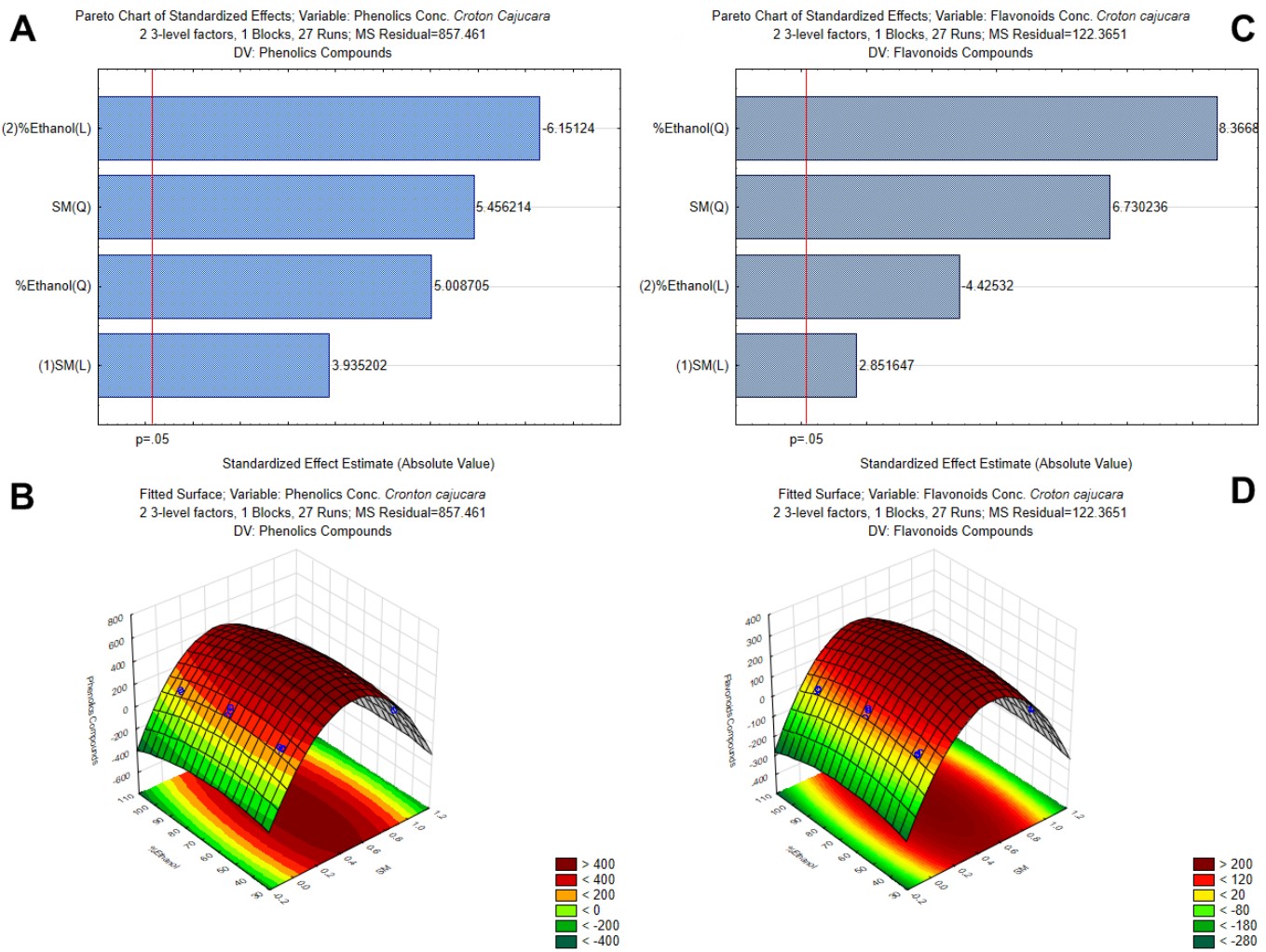

**Figure 2.** Surface response plots and Pareto diagrams showing the effect of the variables amount of plant biomass of *C. cajucara* and concentration of ethanol as solvent extractor (mass:solvent ratio) to evaluate the content of phenolic compounds (**A**,**B**) and flavonoids in the obtained extracts (**C**,**D**).

The critical values obtained through the desirability response profile of the test established a relationship between the predicted responses on one or more dependent variables and the desirability of the responses. This is relevant to indicate the levels of predictor variables capable of generating more desirable predicted responses in the dependent variables. Hence, a fourth test was applied to evaluate the predicted and observed values in order to demonstrate the efficiency of the optimization model adopted in relation to the results displayed by the extracts obtained through the performed experiments.

Based on the fourth test, the antioxidant activity of these extracts was monitored to assess the efficiency of the optimization model adopted to recover total phenolic compounds and flavonoids. Thus, Figures 3–5 depict the comparison between *D. monetaria* and *C. cajucara* extracts related to the antioxidant capacity determined by total antioxidant capacity, DPPH free radical scavenging, and ferric reduction power assays. Predicted and observed values are shown using surface response plots and Pareto diagrams.

Concerning total antioxidant activity, data showed a significant linear response for solvent (% EtOH) and mass (SM) variables in *D. monetaria* extracts. However, all factors on the linear and quadratic axis were significant in the antioxidant analysis of *C. cajucara* extracts (Figure 3). Concerning total antioxidant activity, results showed a significant linear response for solvent and mass variables in *D. monetaria* extracts. All factors on the linear and quadratic axis were significant for the *C. cajucara* extract's antioxidant analysis (Figure 3). On the other hand, no significant effect of the variables was shown by DPPH free

radical scavenging assay with *D. monetaria* extracts, whereas *C. cajucara* extracts exhibited a significant result only for the mass factor as depicted by the Pareto diagram (Figure 4). Towards reducing power, *C cajucara* extracts were significant in all factors and axes and more expressive regarding the % EtOH factor. Relative to *D. monetaria*, results were just significant in the linear response to the % EtOH factor (Figure 5).

Figures 3–5 show that *C. cajucara* extracts displayed greater statistical significance related to the mass and % EtOH factors in both linear and quadratic axes. Nonetheless, the %EtOH factor was the most significant influence on the recovery of total phenolics and flavonoids and in the antioxidant activity determination of extracts compared to the BHT standard. Overall, the optimal conditions for the extraction process using *C. cajucara* were 0.05 g of biomass and 70% ethanol, while for *D. monetaria* were 0.05 g of biomass and solvent concentrations at 40 and 70%.

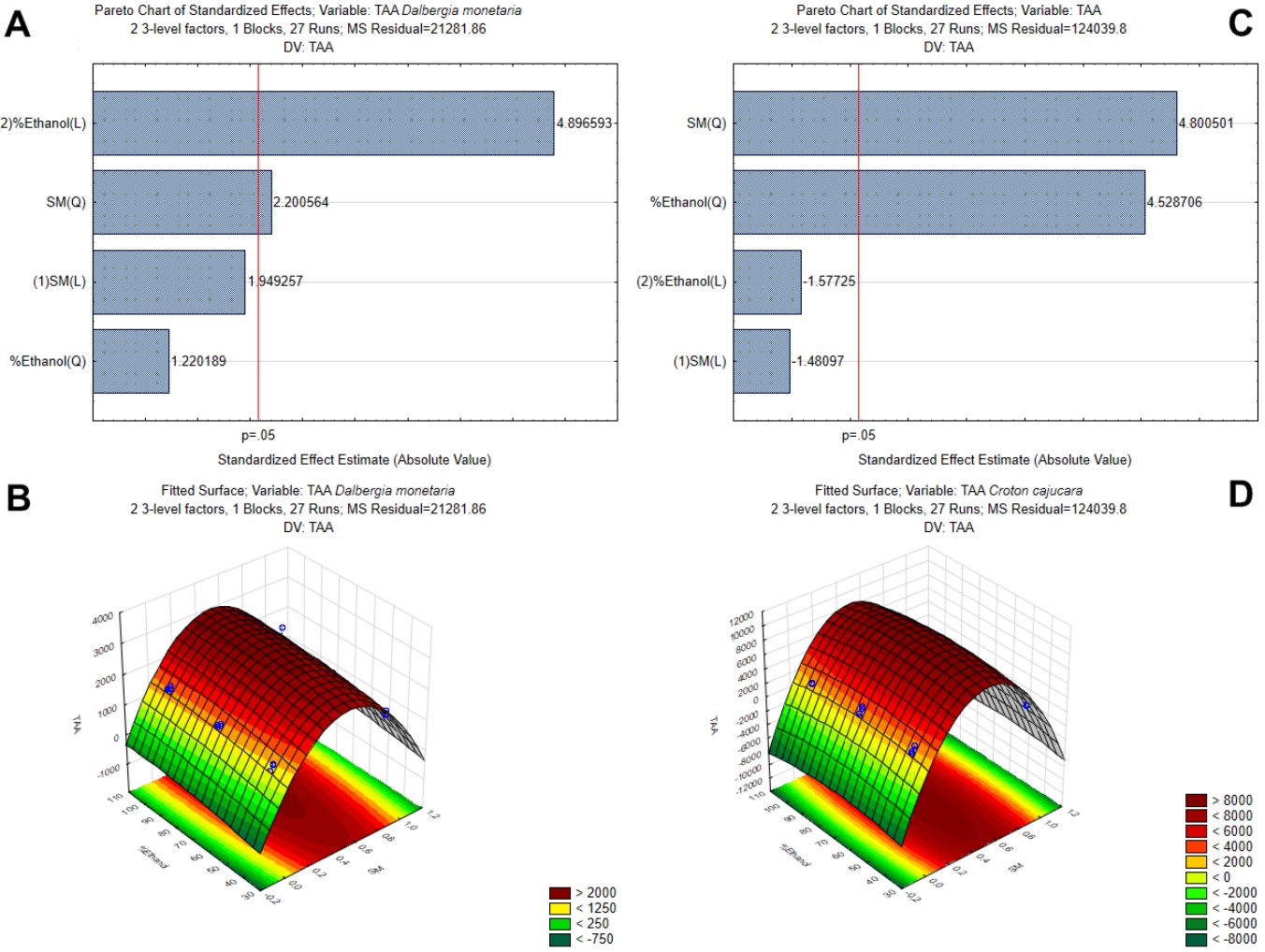

**Figure 3.** Surface response plots and Pareto diagrams showing the effect of mass and ethanol concentration variables on total antioxidant capacity by comparing *D. monetaria* (**A**,**B**) and *C. cajucara* extracts (**C**,**D**).

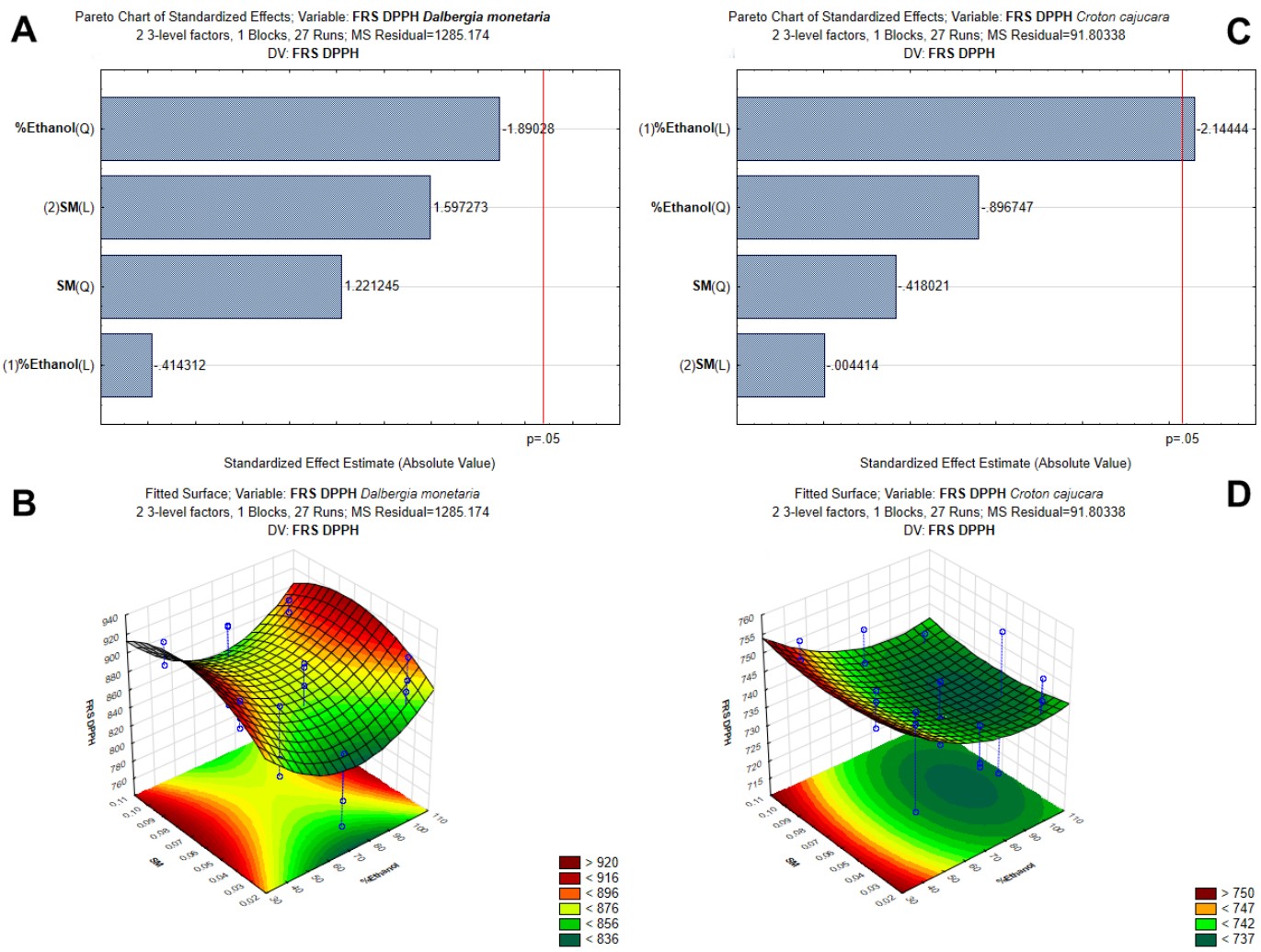

**Figure 4.** Surface response plots and Pareto diagrams showing the effect of mass and ethanol concentration variables on antioxidant capacity by DPPH free radical scavenging assay by comparing *D. monetaria* (**A**,**B**) and *C. cajucara* extracts (**C**,**D**).

These results show the important role of applying a statistical design to optimize extraction conditions and evaluate the response of several factors and their interactions by identifying the most influential factor during this process. Therefore, a statistical design can provide data on extraction to enhance the recovery of bioactive compounds to make better use of biomasses [52,57,60]. Thus, studies have shown the efficiency of applying statistical tools to improve the recovery of several compounds from plant biomasses, mainly polyphenols, aiming at their technological application in different areas, like as food [57], pharmaceuticals [62,63], cosmetics [62] and the incorporation of natural antioxidants in bioactive packaging to preserve and increase the food shelf-life [8,64,65].

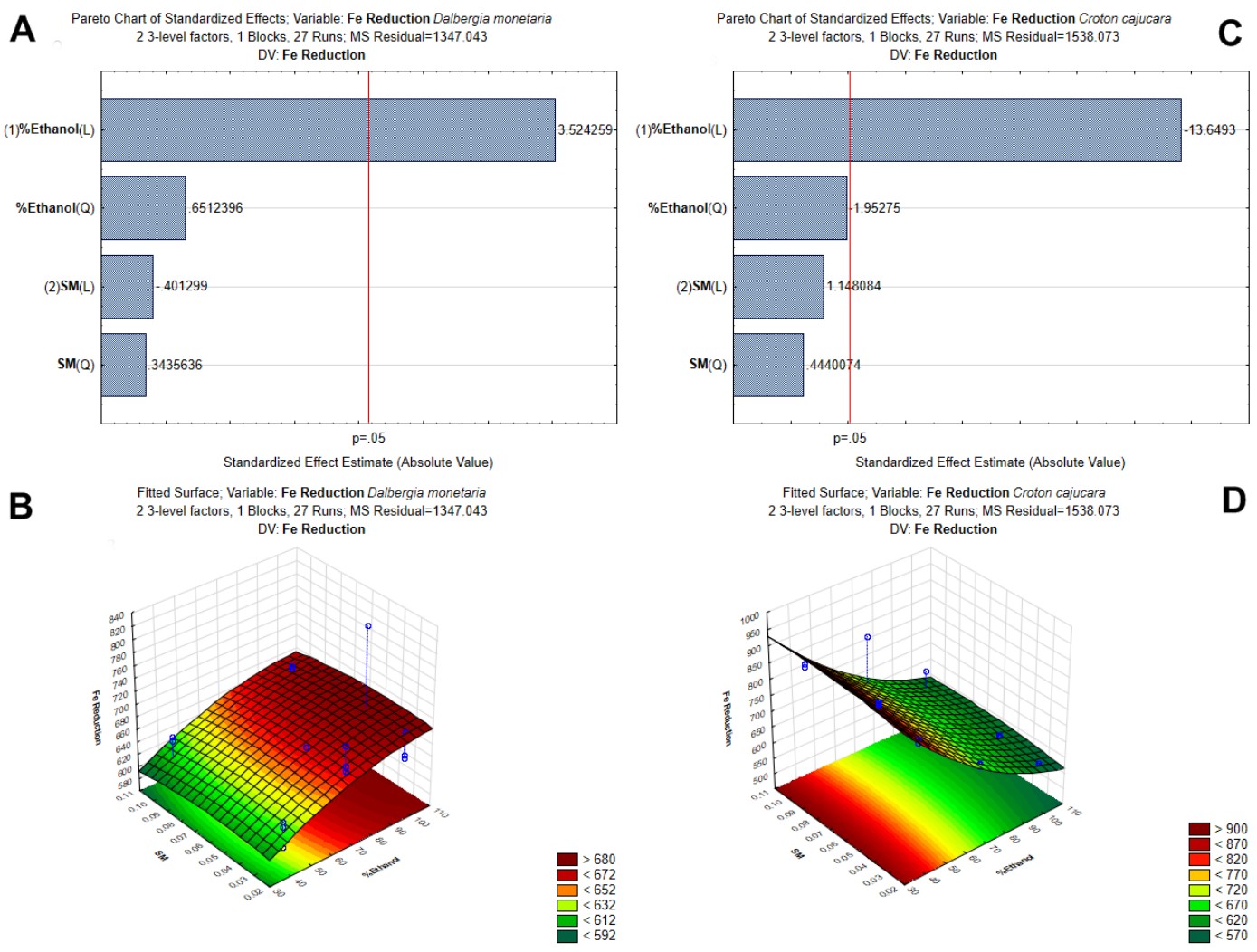

**Figure 5.** Surface response plots and Pareto diagrams showing the effect of mass and ethanol concentration variables on antioxidant activity determined by ferric reduction power assay by comparing *D. monetaria* (**A**,**B**) and *C. cajucara* extracts (**C**,**D**).

### 3.2. UPLC/DAD Analysis of Polyphenols of D. monetaria and C. cajucara Extracts

The extracts selected based on the best extraction conditions were submitted to spectrophotometric assays to identify groups of organic compounds of industrial interest, mainly polyphenols, and to monitor the antioxidant potential of these extracts. Based on this choice, the hydroethanolic extracts obtained were analyzed by UPLC/DAD to identify and quantify the polyphenolic groups by comparing retention times (RT) and UV spectra of the extract and external standard peaks, applying gallic acid and quercetin calibration curves. Analysis of the extract chemical composition revealed nine main compounds, whose UV and RT spectra were similar to those exhibited by standards, such as catechin, (-)-epigallocatechin gallate, rutin, and quercetin. Phytochemical analysis of *D. monetaria* and *C. cajucara* extracts showed the presence of simple phenolic compounds and flavonoids. Table 2 displays the identification of the compounds detected in the extracts obtained under optimal extractive conditions.

**Table 2.** Identification of polyphenols by comparing retention time and UV spectrum of peaks of *D. monetaria* and *C. cajucara* extracts with external standards. Identification was performed by determining the absorption at 280 nm for simple phenolic compounds and at 340 nm for flavonoids based on chromatographic analysis.

| Compound | Retention Times (min) | Linear Range (µg/mL) | UV (nm) | $R^2$ |
|---|---|---|---|---|
| Gallic acid | 1.71 | 1.5–50 | 271 | 0.9945 |
| Chlorogenic acid | 6.47 | 1.5–50 | 326 | 0.9973 |
| Catechin | 7.19 | 1.5–50 | 278 | 0.9993 |
| (-)-Epigallocatechin gallate | 8.18 | 1.5–50 | 275 | 0.9989 |
| Rutin | 9.20 | 1.81–100 | 275–354 | 0.9976 |
| Hyperoside | 9.24 | 1.81–100 | 256–354 | 0.9995 |
| Quercetin | 16.03 | 1.5–50 | 255–371 | 0.9989 |
| Apigenin | 18.51 | 1.5–50 | 266–337 | 0.9989 |
| Kaempferol | 18.85 | 1.5–50 | 263–367 | 0.9984 |

Figure 6 depicts the analysis of *D. monetaria* and *C. cajucara* extracts, revealing the presence of chromatographic peaks closely related to the polyphenolic compounds used as comparative standards. This analysis was established by correlating the maximum UV absorption spectra and retention times between the target sample and the corresponding external standard peaks, as shown in Table 2. This methodology, adapted from Morais et al. [66], also allowed the identification of flavonoids and other phenolic compounds in both analyzed extracts.

Table 3 displays the quantification of polyphenol contents in *D. monetaria* and *C. cajucara* extracts analyzed by UPLC-DAD through an analysis applying two external calibration curves, one with gallic acid and the other with quercetin. The quantification analysis indicated substantial differences in the concentrations of phenolic compounds determined between the two studied species. The results showed a higher content of polyphenols in the *D. monetaria* extract than in *C. cajucara*. Notwithstanding, it was possible to quantify more peaks by UPLC-DAD in the *C. cajucara* extract, which are probably related to phenolic compounds present in its phytochemical composition.

**Table 3.** Quantification of phenolic compounds in *D. monetaria* and *C. cajucara* hydroethanolic extracts expressed as in equivalents of mg of corresponding standard/g of extract according to the most similar phenolic standard spectrum.

| | *Dalbergia monetaria* | | | | *Croton cajucara* | | |
|---|---|---|---|---|---|---|---|
| Peak | RT (min) | Concentration (µg/mL) | UV-$\lambda_{max}$ (min) | Peak | RT (min) | Concentration (µg/mL) | UV-$\lambda_{max}$ (min) |
| 1 | 0.807 | 51.88 | 265 [A] | 1 | 0.651 | 51.72 | 274 [A] |
| 2 | 3.094 | 468.38 | 289/477 [A] | 2 | 5.750 | 5.35 | 335/271 [A] |
| 3 | 20.756 | 119.86 | 332/286 [B] | 3 | 6.286 | 3.53 | 256/349 [A] |
| 4 | 21.960 | 138.94 | 330/662 [B] | 4 | 6.763 | 6.08 | 265/229/346 [A] |
| 5 | 23.236 | 74.76 | 331/491 [B] | 5 | 6.863 | 2.27 | 254/353/232 [A] |
| 6 | 30.737 | 6.77 | 331/491 [B] | 6 | 7.047 | 4.58 | 254/337 [A] |
| | | | | 7 | 7.696 | 2.94 | 265/240/342 [A] |
| | | | | 8 | 7.871 | 3.63 | 254/352 [A] |
| | | | | 9 | 8.043 | 4.98 | 259/346 [A] |
| | | | | 10 | 15.872 | 0.39 | 246/349 [B] |
| | | | | 11 | 19.837 | 2.59 | 267/347 [B] |

[A] mg of gallic acid/g of extract. [B]: mg of quercetin/g of extract. RT: retention time.

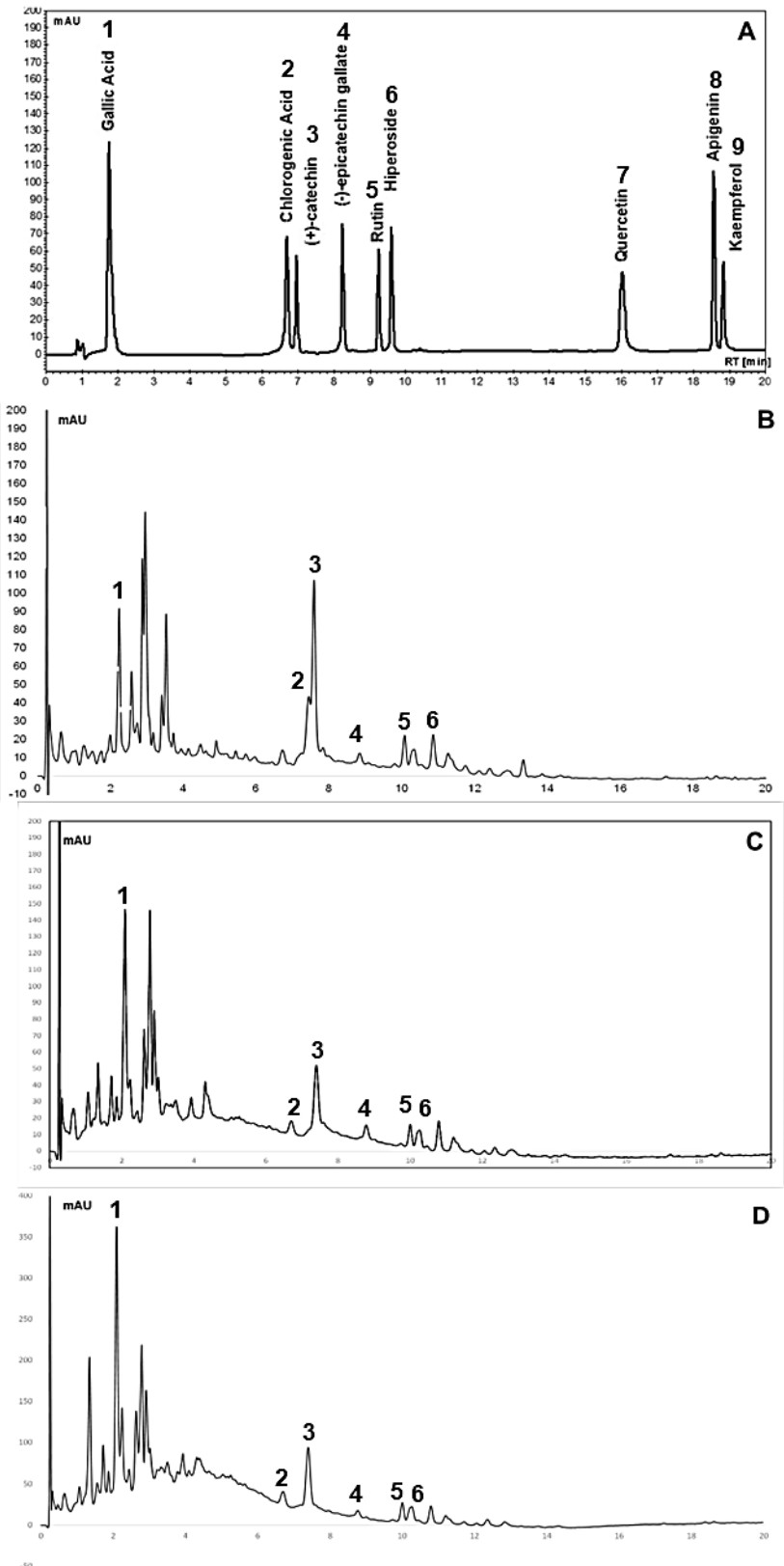

**Figure 6.** Profile of phenolic compounds from the UPLC-DAD analysis of the hydroethanolic extracts of *D. monetaria* and *C. cajucara* corresponding to the best results obtained according to the extractive conditions of the statistical design. (**A**). Chromatogram of standard phenolic compounds. (**B**). *C. cajucara* extract and main peaks (Extractive conditions: 0.05 g and 70% ethanol). (**C**). *D. monetaria* extract and main peaks (Extractive conditions: 0.05 g and 40% ethanol). (**D**). *D. monetaria* extract and main peaks (Extractive conditions: 0.05 g and 70% ethanol).

Although further studies are required, the chromatographic analysis is yet another indication of the importance of implementing a design of experiments strategy to enhance yields in various extractive processes. In this context, studies have reported the application of the response surface methodology as an experimental approach to determine the effect of variables in an extractive process aiming to recover target inputs [59,67,68].

### 3.3. Lipid Peroxidation Inhibition of D. monetaria and C. cajucara Extracts

The ability of the extracts to prevent lipid peroxidation was assessed by quantifying AAPH-induced TBARS in egg yolk homogenate after incubation of the extracts in this lipid-rich medium. The effect of different extract concentrations on lipid peroxidation is shown in Figure 7. Concentrations of *D. monetaria* and *C. cajucara* extracts ranging from 25 to 250 μg/mL were able to reduce AAPH-induced lipid peroxidation with similar values to those displayed by BHT, used as a standard antioxidant, which is widely used in food preservation. The *D. monetaria* extract was more effective than the *C. cajucara* extract in delaying the formation of reactive substances.

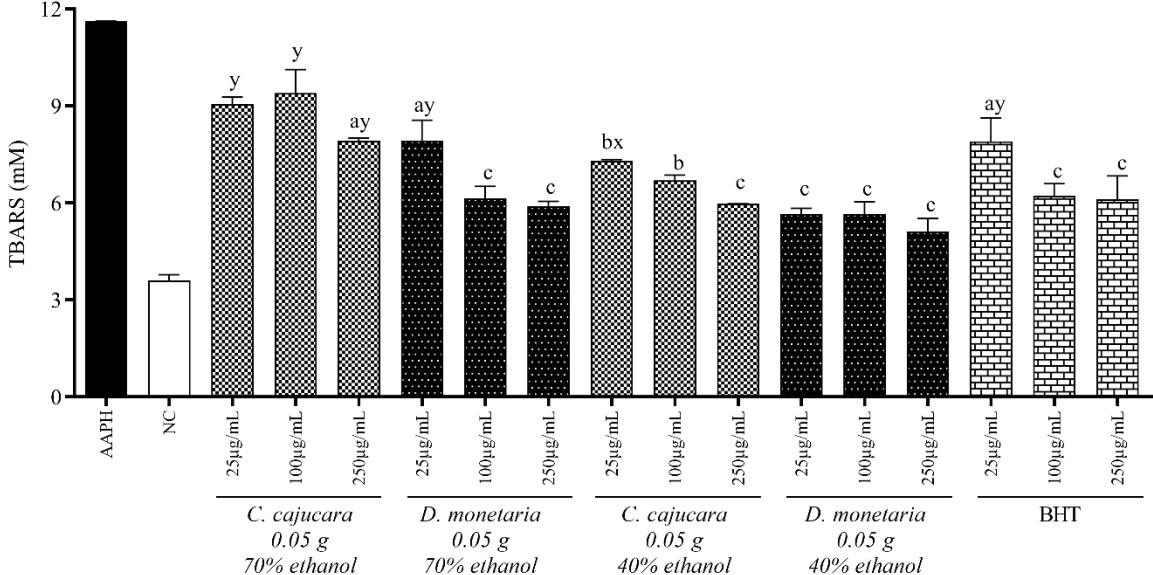

**Figure 7.** Effect of *D. monetaria* and *C. cajucara* hydroethanolic extracts on the inhibition of lipid peroxidation in egg yolk homogenate as a lipid-rich substrate after induction of lipid peroxidation by AAPH. Lipid peroxidation was quantified by thiobarbituric acid reactive species (TBARS). Values are expressed as mean ± standard deviation. One-way ANOVA was used, followed by a post hoc Tukey test. a = $p < 0.05$; b = $<0.005$; c = $p < 0.0001$ compared to AAPH. x = $p < 0.05$; y = $p < 0.005$; z = $p < 0.001$ compared to NC. NC = negative control.

Regarding the *C. cajucara* extract, its ability to reduce lipid peroxidation was similar to that observed for BHT, although the extract obtained under the conditions of 0.05 g and 70% ethanol showed better results. On the other hand, the antioxidant effect of *D. monetaria* extracts was more efficient in reducing lipid peroxidation, mainly the extract from the extraction under the conditions of 0.05 g and 40% ethanol. The extract obtained with 70% ethanol was efficient only at concentrations of 100 and 250 μg/mL.

The results observed concerning the ability to prevent lipid peroxidation are supported by the phytochemical composition that showed a significant content of phenolic compounds applying some assays for their quantification. The chromatographic analysis indicated significant values of polyphenols in *D. monetaria* extracts compared to the content of these compounds determined in *C. cajucara* extracts. Overall, phytocomposition is closely associated with the antioxidant effect. Although further assays are required, data indicate the potential of both extracts as a food preservative compared to the BHT results.

Lipid peroxidation is a critical factor because it is closely related to extensive food spoilage, which is attributed to unpleasant flavors and odors to the product, making it unfit for consumption by modifying the nutritional quality and formation of potentially toxic compounds [69,70]. This process is responsible for forming volatile aromatic compounds, such as ketones and aldehydes, known as off-flavors [71,72]. Malondialdehyde is the most abundant aldehyde produced during secondary lipid oxidation and is commonly used as a marker to determine the presence of oxidized lipids in foods, which implies an increase in TBARS content. Hence, polyphenol-rich extracts are correlated by their ability to scavenge free radicals in alleviating spoilage caused by TBARS in foods [73]. Therefore, the results observed regarding the antioxidant effect and decrease in lipid peroxidation of *D. monetaria* and *C. cajucara* extracts are promising, considering the scientific interest in natural bioactive compounds to replace synthetic food additives and meet industrial and consumer demands on food safety, quality, and shelf-life extension [11,44,74].

## 4. Conclusions

Based on the optimized results, *D. monetaria* and *C. cajucara* extracts exhibited positive responses regarding the total polyphenol quantification, confirmed by UPLC analysis. The *C. cajucara* extract displayed more significant responses regarding the antioxidant capacity compared to the BHT standard under conditions of 0.05 g of vegetable biomass:solvent and 70% ethanol as extracting agent. Significant results were also observed under the same biomass condition in the extraction using 40% of ethanol. The antioxidant effect was evidenced by a significant decrease in TBARS content, evaluated through the lipoperoxidation inhibition assay. Although *D. monetaria* extracts showed significant results under the same applied conditions, a better response to the adopted optimization model was observed with *C. cajucara* extracts. Nevertheless, further studies are required to evaluate other variables in order to obtain a more robust response for an optimized polyphenol extraction. This is pivotal considering the significant antioxidant activity of the analyzed species extracts and their potential application in several industrial sectors.

**Author Contributions:** Conceptualization, G.A.-S. and L.G.d.S.S.; methodology, V.A.P.d.A., G.A.-S. and L.G.d.S.S.; formal analysis, V.A.P.d.A., J.R.D.d.L., J.A.L., G.A.-S. and L.G.d.S.S.; Investigation, V.A.P.d.A., N.S.d.S.e.S., M.P.P. and D.C.d.S.; resources, G.A.-S. and L.G.d.S.S.; data curation, J.P.B., J.R.D.d.L., J.A.L., G.A.-S. and L.G.d.S.S.; writing-original draft preparation, V.A.P.d.A., J.R.D.d.L. and J.A.L.; writing—review and editing, J.A.L., J.R.D.d.L., G.A.-S. and L.G.d.S.S.; supervision, G.A.-S., L.G.d.S.S. and J.P.B.; project administration, G.A.-S. and L.G.d.S.S.; funding acquisition, G.A.-S. and L.G.d.S.S. All authors have read and agreed to the published version of the manuscript.

**Funding:** This research was funded by JBS Fund for the Amazon (Ordinance No. 348/2021-UEAP), Research Support Foundation of the State of Amapá-FAPEAP (Grant No. FAPEAP/Decit/SCTIE/MS/ SESA-AP/CNPq No 003/2020.

**Institutional Review Board Statement:** Not applicable.

**Informed Consent Statement:** Not applicable.

**Data Availability Statement:** Not applicable.

**Acknowledgments:** The authors would like to thank the Coordination for the Improvement of Higher Education Personnel (CAPES) (Finance Code 001) and the Foundation of the State of Amapá-FAPEAP (Grant No. FAPEAP/Decit/SCTIE/MS/SESA-AP/CNPq No 003/2020.

**Conflicts of Interest:** The authors declare no conflict of interest.

**Sample Availability:** Not available.

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
