# Peer review of "Optimization of Polyphenol Extraction with Potential Application as Natural Food Preservatives from Brazilian Amazonian Species Dalbergia monetaria and Croton cajucara"

_processes, doi:10.3390/pr11030669_

Round 1

Reviewer 1 Report

I have gone through the manuscript submitted to Processes. Authors have investigated the antioxidant activity of extracts from two species, Dalbergia monetaria and Croton cajucara, colelcted from Brazialian Amazon forests. The experimetns have been nicely designed and conducted meticulosly. the data has been organized and presented well. I have a few queries/ suggestions:

1. Why did authors choose these two species? A robust explanation is required in view of the high endemism / utilization of Amazonian plant species.

2. In my opinion, the phrase "optimized antioxidant" seems out of place. It appears authors wish to explain that they have optimized the antioxidant activity tests using polyphenols as criteria! Accordingly, changes may be made in the title and abstract.

3. What were the sources of external reference compounds used?

4. Authors must spell the acronyms/abbreviations at the first instance.

5. Why was lipid peroxidation assay undertaken? How was it related to optimization of antioxidant activity of extracts?

Reviewer 2 Report

This manuscript is about hydroalcoholic extraction of polyphenols of two amazonian plants, and I have some considerations about it:

- In material and methods, I did not understand very well the idea of gravimetric analysis. Would be the total extractive with solvent mixture after dried? Was it the response paramter in experimental design instead of polyphenols concentration? Explain please

- In the item Experimental design, please adjust the information about mass/solvent. I think it is 1/10, 1/20...

- I same item, inform the conditions which was maintained constant. What temperature, granulometry...

- Improve the resolution in all figures. Try to save in tiff format

- In Figures 3,4,5, it is not clear what subfigure is veronica or sacaca, please include letters A, B, C and D to better identification.
